# Infections in Glucose-6-Phosphate Dehydrogenase G6PD-Deficient Patients; Predictors for Infection-Related Mortalities and Treatment Outcomes

**DOI:** 10.3390/antibiotics12030494

**Published:** 2023-03-01

**Authors:** Diaa Alrahmany, Ahmed F. Omar, Wael Hafez, Sara Albaloshi, Gehan Harb, Islam M. Ghazi

**Affiliations:** 1Pharmaceutical Care Department, Directorate General of Medical Supplies—MOH, Muscat 112-393, Oman; 2General Medicine Department, Suhar Hospital, Suhar 311-49, Oman; 3NMC Royal Hospital, Abu Dhabi P.O. Box 764659, United Arab Emirates; 4The Medical Research Division, Department of Internal Medicine, The National Research Centre, Cairo 12622, Egypt; 5GH Statistics, Cairo 11511, Egypt; 6Arnold and Marie Schwartz College of Pharmacy and Health Sciences, Long Island University, Brooklyn, NY 11201, USA

**Keywords:** G6PD deficiency, bacterial infections, mortality rates, length of stay, hospital-acquired infections, prior exposure to antibiotics, monotherapy, combined therapy, history of infections, polymicrobial infections

## Abstract

Disturbances in the count or maturity of blood cells weaken their microbial defensive capacity and render them more susceptible to infections. Glucose-6-phosphate deficient patients are affected by a genetic disease that affects cell integrity with increased liability to infections and death. We aimed to investigate the risk factors for infection mortality in this patient population. We retrospectively examined the records of G6PD adult patients with confirmed infections and collected data related to demographics, infections (pathogens, types, and treatment regimens) in addition to mortality and length of stay outcomes. Data were statistically analyzed using R Programming language to identify contributing factors to mortality and treatment regimens association with outcomes. Records of 202 unique patients over 5 years were included, corresponding to 379 microbiologically and clinically confirmed infections. Patients > 60 years [*p* = 0.001, OR: 5.6], number of comorbidities 4 (2–5) [*p* < 0.001, OR: 1.8], patients needed blood transfusion [*p* = 0.003, OR: 4.3]. Respiratory tract infections [*p* = 0.037, OR: 2.28], HAIs [*p* = 0.002, OR: 3.9], polymicrobial infections [*p* = 0.001, OR: 10.9], and concurrent infection Gram-negative [*p* < 0.001, OR: 7.1] were significant contributors to 28-day mortality. The history of exposure to many antimicrobial classes contributed significantly to deaths, including β-lactam/β-lactamase [*p* = 0.002, OR: 2.5], macrolides [*p* = 0.001, OR: 3.34], and β-lactams [*p* = 0.012, OR: 2.0]. G6PD patients are a unique population that is more vulnerable to infections. Prompt and appropriate antimicrobial therapy is warranted to combat infections. A strict application of stewardship principles (disinfection, shortening the length of stay, and controlling comorbid conditions) may be beneficial for this population. Finally, awareness of the special needs of this patient group may improve treatment outcomes.

## 1. Introduction

The non-specific innate immune response is the frontline defense against infections; releasing a sufficient number of different types of the innate immune system’s myeloid cells (monocytes, macrophages, dendritic cells (DCs), neutrophils, eosinophils, basophils, mast cells, and platelets) to achieve integrated highly specialized functions that allow them to recognize and combat pathogens [1,2]. Any disturbance in the count or maturity of these cells caused by genetic blood diseases significantly impacts the body’s ability to resist bacterial or viral infections [3,4].

Glucose-6-phosphate dehydrogenase (G6PD) is an essential enzyme for the maintenance of blood cell integrity [5]. The G6PD enzyme stimulates the reduction in nicotinamide adenine dinucleotide phosphate (NADP) in the pentose phosphate pathway (PPP) to generate NADPH; the latter is a substrate for NADPH oxidase responsible for regenerating the antioxidant glutathione that protects the blood cells against oxidative stress. The deficiency of G6PD leads to premature loss of cell integrity under the stress of sulfa-containing drugs, certain foods, and systemic infections, which explains why many studies suggested G6PD deficiency to be a significant predictor of hospitalization and severe infections [3,4,6,7].

Several studies identified a slew of life-threatening infections in G6PD-deficient patients, with bacteremia, respiratory, cerebrospinal, and urinary tract infections being the most common. Gram-negative bacteria were the predominant cause of these infections, followed by Gram-positive bacteria and, to a lesser extent, a variety of fungi and viruses [7,8,9].

The inability of the immune system to resist infectious diseases, combined with other factors that could elevate the risks of infections result in high rates of infection-related deaths.

The scarcity of localized G6PD enzyme deficiency cases around the world challenges researchers to track the accompanying clinical changes in a sufficient number of cases. In our community—due to the high prevalence of consanguineous marriage—a large number of G6PD enzyme deficiency cases is registered, which may reach 25% in males and 10% in females, providing a unique opportunity to study these cases in depth that may not be possible in other societies [10]. In a previous study by the same research group, we monitored the pattern of infection in an adequate number of G6PD patients, allowing us to vividly describe the causes and risk factors for nosocomial infections and infections with multi-drug resistant (MDR) bacteria [7]. Therefore, we found it useful to track the predictors that may be associated with high mortality rates in the same patient population that primarily suffers from an inferior immune response. Additionally, as part of an initiative to guide antimicrobial stewardship programs (ASP), we evaluated the antimicrobial treatment regimens used to determine which are most related to recovery rates and successful treatment outcomes.

## 2. Method

### 2.1. Study Population

This study included G6PD-deficient genetically tested adult patients (>18 years) with laboratory-confirmed microbial infections who were admitted to our tertiary care hospital (Suhar Hospital, Oman) between 1 January 2017, and 31 December 2021. The relevant data were collected from the patient’s electronic medical records after obtaining ethical approval from the Ministry of Health’s Research and Ethical Review Committee.

We examined the patient’s age, gender, clinical symptoms of infection (to exclude patients with colonization), existing comorbid conditions, diabetes mellitus (DM), chronic kidney disease (CKD), active malignancy, immuno-suppression, chronic cardiac diseases (CCD), chronic respiratory disease (CRD), exposure to invasive procedures (endotracheal tube insertion, urinary catheterization, wound debridement, venous catheterization, lumbar puncture or similar procedures) during admission, 90-day prior exposure to any surgery, and 90-day history of infections. Hospitalization details included diagnosis at admission, discharge status, length of stay (LOS), and admission ward.

### 2.2. Definitions

Chronic kidney disease; an estimated glomerular filtration rate (eGFR) of <60 mL/min/1.73 m^2^. Chronic respiratory diseases; chronic obstructive pulmonary disease, bronchiectasis, cystic fibrosis, and asthma. Chronic cardiac diseases include heart failure, hypertension, rheumatic heart disease, cardiomyopathy, arrhythmias, congenital heart disease, valvular heart disease, aortic aneurysms, peripheral artery disease, thromboembolic disease, and venous thrombosis. Immunocompromised patients are those receiving T-cell immunosuppressants, tumor necrosis factor (TNF) blockers, specific monoclonal antibodies, or corticosteroids at a dose of 0.3 mg/kg/day of prednisone for at least 72 h before the index infection. A critical care stay is considered if the patient is admitted to the Intensive Care Unit (ICU), Cardiac Care Unit (CCU), or Burn unit (BU) for more than 24 h. Infection-related death was considered if the symptomatic patient had a positive culture and died before the resolution of infection symptoms during the same hospitalization. The following criteria demonstrate infection resolution: subsidence of presenting symptoms, normal inflammatory laboratory values, or negative culture of the same source as the original index infection.

### 2.3. Microbiological Details

Laboratory-confirmed microbiological cultures, infection sites, specimen type, susceptibility pattern, resistance phenotype, prior infections, and concurrent infections. Only the first episode was selected for patients with identical cultures in terms of the organism, sample source, and susceptibility pattern; identical cultures isolated within 30 days for the same patient are considered unique [11]. The antimicrobial treatment using a single antibiotic is considered monotherapy, while combined therapy is the use of 2 or more antibiotics with antimicrobial effects toward the causative pathogen. Hospital-acquisition: infection occurred ≥72 h of the admission date; all other episodes were considered community-acquired infections CAI [12]. Cultures correspond to patients with no hospital ID. Patients with positive cultures who were not admitted, patients who died before receiving a single dose of antibiotics, and pediatric patients (<18 years) were excluded.

### 2.4. Statistical Analysis

The data were analyzed using R software statistical programming language, (R Foundation for Statistical Computing platform). Median and interquartile ranges (IQR) were used to describe numerical data and analyzed using linear regression analysis after the normality was tested using Shapiro–Wilk normality test. Categorical data were analyzed using binary logistic regression and expressed using *p* values, odds ratios (OR), and confidence intervals (CI). All tests were two-sided; *p*-values < 0.05 were considered significant, at a 95% confidence level.

Binary logistic analyses were used to direct the correlation between all variables and dependent variables. To quantify the cumulative effect, all variables with a *p* ≤ 0.2 in the bivariate analysis were included in a multivariate regression model. We studied the statistical relation between antibiotic regimens and 14, 28, and overall all-cause mortality as the primary outcomes, in addition to the length of hospital stays (LOS) as a secondary outcome.

## 3. Results

The records of 3334 registered G6PD-deficient patients between 1 January 2017, and 31 December 2021, were reviewed; 2512 patients were excluded because they were <18 years when they had a laboratory-documented bacterial infection during the study period, while 620 other adult patients were excluded as they did not have any microbiological cultures or hospitalization details. The remaining 202 patients’ records were examined over 5 years, and 379 microbiological cultures corresponding to hospital admissions were recorded and studied. See Figure 1.

### 3.1. Patients’ Demographics

The study participants were 68% men. The patients were equally distributed around the median (IQR) age at the admission of the overall sample of 60 (41–77). A total of 89% of the patients were having at least one underlying comorbidity, mainly chronic cardiac diseases (CCD) 74%, diabetes (DM) 67%, chronic kidney disease (CKD) 60%, and chronic respiratory disease (CRD) 18%. The median (IQR) length of stay is 12 (5–31).

The need for critical care admission occurred in 31% of the patients, while 75% of the cohort were subjected to an invasive procedure, and 56% needed a blood transfusion during admission. Table 1 shows the details of the patient’s demographics.

Skin and soft tissue infections accounted for 27% of infections, urinary tract infections 25%, respiratory tract infections 24%, and bacteremia 23%. Infections due to Gram-negative bacteria predominate in 60% of cases, followed by Gram-positive 28%, fungal 8%, and SARS-CoV-19 infections 4%. Resistant bacterial phenotypes caused 40% of the infections; meanwhile, 44% were hospital-acquired infections (HAIs). Polymicrobial infections occurred in 59% of the cases, mainly with Gram-negative 45%, Gram-positive 28%, fungi 16%, and 2% with SARS-CoV-19.

Almost 23% of the patients had a 90-day history of infection, of which Gram-negative bacteria was 13%, 9% Gram-positive, SARS-CoV-19 5%, and Fungi 1%. While 42% of patients were exposed to prior use of antibiotics, 66% of them used Cephalosporins, β-lactams 44%, quinolones 41%, β-lactam/β-lactamase 37%, and others. See Table 1.

### 3.2. Univariate Analysis

#### 3.2.1. Crude, In-Hospital Mortality (105, 28%)

Age at admission possessed high statistical significance, patients > 60 years died at an 8-fold greater rate than those below 60 years [*p* < 0.001, OR: 8.2]. Prolonged LOS (>14 days) was a significant predictor of mortality [*p* < 0.001, OR: 4.6]. Patients presented with an infectious disease at admission died at a 3-fold greater rate compared to those admitted with other diagnoses [*p* < 0.001, OR: 2.98]. Meanwhile, patients admitted to critical care areas died at a 5-fold greater rate than those admitted to general wards [*p* < 0.001, OR: 5.1].

The existence of any comorbidities [*p* = 0.005, OR: 17.8], as well as the number of comorbidities [*p* < 0.001, OR: 2.0], DM [*p* = 0.001, OR: 2.5], CKD [*p* < 0.001, OR: 4.6], CCD [*p* < 0.001, OR: 5.8] and CRD [*p* < 0.001, OR: 6.3], were all highly significant predictors of crude mortality. Bacteremia [*p* = 0.040, OR: 1.7], respiratory tract [*p* = 0.001, OR: 2.4], and MDR-related infections [*p* = 0.005, OR: 2.19] were all significant contributors to crude mortality. HAIs contributed to a 4-fold increase in mortality compared to community-acquired infections (CAIs) [*p* < 0.001, OR: 4.4].

History of exposure to β-lactam/β-lactamase [*p* = 0.002, OR: 2.49], macrolides [*p* = 0.001, OR: 3.34], and β-lactams [*p* = 0.012, OR: 2.0] showed statistically significant high odds of deaths. Post-covid and polymicrobial infections significantly contributed to crude mortality [*p* < 0.001, OR: 17.5], and [*p* < 0.001, OR: 4.8], respectively. See Table 2 for the detailed statistical analysis.

##### 3.2.2. 14-Day Mortality (27, 26% of Overall Mortality)

Male gender was a significant predictor for early-onset mortality [*p* = 0.036, OR: 3.7], while patients > 60 years were more susceptible to death [*p* = 0.002, OR: 4.9]. Patients subjected to invasive procedures during admission were 4-fold more liable to early deaths [*p* = 0.046, OR: 4.4], the same is copied with patients who had bacteremia [*p* = 0.002, OR: 3.4]. Infections with Gram-positive pathogens, and CAIs were significant predictors of 14-day mortality, [*p* = 0.020, OR: 2.6], and [*p* = 0.009, OR: 3.8], respectively. Concurrent infection with SARS-CoV-19 during the same infection episode was a significant predictor of early-onset death [*p* = 0.003, OR: 10.9]), respectively. See Table 3.

##### 3.2.3. 28-Day Mortality (30, 29% of Overall Mortality)

Patients > 60 years died 6-fold more than those below 60 [*p* = 0.001, OR: 5.6]. The number of comorbidities 4 (2–5) was a significant contributor to 28-day mortality [*p* < 0.001, OR: 1.8] mainly active malignancy (*p* < 0.001, OR: 10.7], and Immunosuppressed [*p* = 0.003, OR: 4.6]. Patients who needed blood transfusion were 4-fold more exposed to death [*p* = 0.003, OR: 4.3]. Respiratory tract infections [*p* = 0.037, OR: 2.28]), HAIs [*p* = 0.002, OR: 3.9], polymicrobial infections [*p* = 0.001, OR: 10.9]), concurrent infection Gram-negative [*p* = 0.000, OR: 7.1], and history of exposure to many antimicrobial classes contributed significantly to deaths, including β-lactam/β-lactamase [*p* = 0.024, OR: 2.6], macrolides [*p* = 0.001, OR: 4.5], and β-lactams [*p* = 0.010, OR: 3.6]. See Table 4.

### 3.3. Multivariate Analysis: All Variables with p ≤ 0.2 in the Univariate Analysis for Each Dependent Variable Were Included in a Multivariate Regression Model

#### 3.3.1. Crude, In-Hospital Mortality

Age (*p* = 0.003), LOS > 14 days (*p* = 0.004), admission with infectious disease (*p* = 0.015), admission to critical care area (*p* = 0.002), exposure to invasive procedures (*p* < 0.001), HAIs (*p* = 0.043), and previous exposure to β-lactam/β-lactamase, were collectively significant contributors to crude mortality.

#### 3.3.2. 14-Day Mortality

Exposure to invasive procedures (*p* < 0.001), exposure to β-lactam/β-lactamase (*p* = 0.013), and prior exposure to aminoglycosides [*p* = 0.020], were collectively significant contributors to 14-day mortality.

#### 3.3.3. 28-Day Mortality

Age > 60 years (*p* = 0.009), CKD (*p* = 0.015), CRD (*p* < 0.001), blood transfusion during admission (*p* = 0.003), HAIs (*p* = 0.021), 90-day previous exposure to cephalosporins (*p* = 0.007),β-lactam/β-lactamase (*p* < 0.001), as well as polymicrobial infection (*p* = 0.024), were collectively significant contributors to 28-day mortality.

### 3.4. Antimicrobial Treatment Regimen and Treatment Outcomes

According to antimicrobial regimens, the cohort (379) was divided into 129 (34%) patients who received combined therapy while 250 (66%) received monotherapy. The antimicrobial regimens during the infection episode were mainly: Cephalosporin (41%), β-lactam/β-lactamase inhibitor (35%), Piperacillin/tazobactam (26%), Quinolones-based therapies (12%), and others. Quinolones and β-lactams based therapies were the only regimens that are significantly related to low odds of overall mortality *p* = 0.00 and *p* = 0.05, respectively. Meanwhile, none of the antimicrobial therapies had a statistically significant relation to either 14-day or 28-day mortality. Colistin and Piperacillin/tazobactam-based therapies were a significant contributor to prolonged LOS *p* = 0.01, and *p* = 0.00, respectively, probably due to their use in complicated cases; meanwhile, other classes were associated with statistically non-significant shorter LOS. See Table 5.

## 4. Discussion

This retrospective analysis aimed to identify the variables that may contribute to the high infection-related mortality rates in G6PD-deficient patients. The focus on the G6PD-deficient patient group originated from the fact that this genetic disease imparts an immunodeficiency status as suggested in the literature, a decrease in the synthesis of reactive oxygen radicals resulting in diminished activation of the NF-κB pathway [13], increased susceptibility to infection due to neutrophil extracellular trap formation/neutrophil elastase translocation malfunction [14], and increased rate of recurrent infections due to impairment of the microbicidal activity of phagocytes [15]. In addition to the assessment of the antimicrobial treatment outcomes, we analyzed 16 variables presumed to contribute to crude, 14-day, and 28-day mortality as primary outcomes; meanwhile, LOS was studied as a secondary outcome.

### 4.1. Predictors of Infection-Related Mortalities

**Age**: It was logical that aging contributes statistically to an increase in crude mortality rates as the deaths of patients > 60 were 6–8 times higher than those among patients who did not exceed this age. This can be attributed to the expected deterioration of health status (98% of them had at least one comorbidity) and immune capacity of the individuals under study, (62%) who had a polymicrobial infection mainly hospital-acquired (43%), bacteremia and respiratory infections 54%), regardless of the presence of other factors that may contribute to the increase in mortality rates [16,17]. In terms of early and medium-onset mortality, ages > 60 years are predisposed to a (5–6) fold increase in mortality compared to ages < 60 years.

**Male gender** died 4-fold earlier compared to females in terms of 14-day and 28-day mortality. G6PD deficiency was found to be more prevalent in infected males and suggested to be a predictor of hospitalization and severe infections and consequently higher mortality rates probably due to the association between sex-linked gene and G6PD leading to predominance of more severe disease in males [18].

**Prolonged LOS** (>14 days) was a significant contributor to crude mortality probably explained by the high liability to HAIs, Hassan, and colleagues reported a similar observation in patients discharged from their hospital, in New Jersey, USA; they concluded that one day increase in LOS increases the probability of acquiring infection by 1.37% and the onset of infection by 9.32 days [19].

**Presenting to the hospital with an infectious disease** was a highly significant predictor of crude mortality. The 3-fold increase in odds of mortalities compared to patients admitted with other diagnoses can be attributed to the fact that a large proportion of these patients (53%) were over the age of 60 and had at least one chronic disease (95%), the most common of which were CCD (82%), diabetes (73%), and CKD (69%). Furthermore, (57%) of these patients were having polymicrobial infections, mainly MDR-related (41%). Chafranska and colleagues identified several independent predictors for mortality in patients presenting to the emergency department with infectious diseases [20]; in the same context, a large-scale retrospective study conducted in Vietnam found sepsis at ICU admission to be associated with the highest in-hospital mortality (37%) [21]. In this study, infection with Gram-positive and SARS-CoV-19 were related to a 3 and 4-fold increase in early-onset mortality, respectively.

**Admission to critical care** contributed to a 5-fold increase in overall mortalities, probably due to the fact that the majority of patients admitted to those areas had a deteriorated health status, (86%) had at least one comorbidity (mainly CCD 84%, DM 68%, and CKD 62%). A total of 80% of them acquired the infection during hospitalization, which was mainly polymicrobial (82%) and Gram-negative-related infections (73%). The patients’ high exposure to blood transfusions (91%) and invasive procedures (95%) as potential sources of infection during their intensive care stay may have also contributed significantly to their high mortality rates. While, several studies disagreed with this result, possibly due to differences in the level of healthcare or the severity of the cases under study [22,23,24], the same finding was reported by others [25,26].

**Exposure to invasive procedures such as** intubation/catheterization/any surgical approach was a significant contributor to early death (14-day), whereas all patients who died ≤28 days were subjected to an invasive procedure during admission. This can be attributed to two hypotheses that appear to be contradictory, but clinically one or both of them can occur. Since it can be assumed that the severity of the case necessitated such procedures, but that they were insufficient to save the patient or that the procedure itself was a source of infection [27,28], this can also be generalized to cases that required blood transfusion, and this was one of the factors that statistically affected 28-day mortalities. This emphasizes the importance of improved risk stratification based on an evidence-based need for these procedures in order to mitigate the boosting effect on mortality and morbidity.

**Underlying comorbid conditions**, particularly DM, CKD, and CCD, had a remarkable statistical impact on crude mortality rates, with death rates exceeding 2–6 times those of the comparable patients. Our findings were in accordance with Simpson and colleagues, who identified a number of chronic illnesses that were associated with higher mortality rates compared to other diseases [26]. In the sample under study, although the patients’ underlying chronic illness did not have a statistical impact on early mortality, it had a clear impact on 28-day deaths. This kind of information provides a valuable guide to identifying high-risk patients who required prompt and intensive intervention.

**Site of infection:** All of the observed infection sites had a statistically significant effect on the crude death rate, most likely due to the high proportion of polymicrobial infections (83%) and MDR bacteria (45%), which were initially a significant contributor to crude mortality in the study cohort. Our results matched several studies where pneumonia, bacteremia, and urosepsis [29,30], as well as MDR-related infections [31], were identified among the most common causes of mortality in hospitalized patients. Our study identified a significant impact of bacteremia on 14-day mortality and respiratory infections on 28-day mortality. Similar findings produced by other researchers [32,33,34] necessitate the identification of modifiable confounders with which we can intervene therapeutically to reduce related mortality.

**HAIs** contributed to 4-fold rises in overall mortality as well as 28-day mortality compared to CAIs, which was common in many earlier studies, they suggest that continuous education and encouraging policies for healthcare professionals to adopt simple infection-control procedures help in reducing HAIs and related costs and mortality [35,36,37]. Infection control measures should be implemented to minimize the spread of hospital-acquired infections in addition to empowering the periodic surveillance of nosocomial infections.

**Polymicrobial infections** were one of the leading causes of crude, 14-day, and 28-day deaths in the study sample, as they had been in many previous studies. Several studies have recorded diverse rates of concurrent-polymicrobial infection-related mortality. This discrepancy may have been a logical effect of the difference in the nature, severity of cases, and underlying comorbid conditions [38,39]. Although a diversity of pathogens, including *Acinetobacter baumannii, Klebsiella pneumonia*, Coagulase-negative Staphylococcus, and candida species, are commonly involved in polymicrobial infections, the strong association between polymicrobial infections and high mortality rates is maintained over all pathogens [40,41,42]. In our cohort, the deteriorating immune status of these patients contributed to the increased exposure to polymicrobial infections as a large proportion of them suffer from chronic diseases (92%), mostly diabetes and heart diseases, in addition to the exposure to invasive procedures (87%) and needed a blood transfusion due to excessive hemolysis triggered by infections (70%).

**Prior exposure to antimicrobials** mainly macrolides had a noticeable influence on overall and 28-day mortality rates, taking together the similar effect of concurrent infections with Gram-positive bacteria, extensive exposure to macrolides upregulates the expression of ermB gene responsible for macrolide, lincosamide, and streptogramin B (MLSB) antimicrobial resistance [43]. In Gram-positive-related infections, almost 45% of isolates were erythromycin-resistant. In the same context, prior exposure to β-lactams was a significant predictor of crude and 28-day mortality, which was supported by many research endeavors that recommended considering previous antimicrobial treatment when initiating an empiric treatment for a current infection due to presumed resistance [44,45], tracking and updating the resistance pattern remains the most effective approach to determine the optimal empiric antimicrobial therapy.

### 4.2. Clinical Outcomes vs. Antimicrobial Treatment

#### Monotherapy vs. Combined

In general, neither combination therapy nor monotherapy treatment appears to have a significant effect on mortality rates or LOS; however, when compared to monotherapy, combined therapy was associated with lower odds of 28-day mortality and a shorter LOS. While some studies replicated the same findings [46,47], others found monotherapy is superior in terms of safety and thus contributes to lower mortalities and shorter LOS [48], while many systemic reviews supported the superiority of combined therapy over monotherapy for treating MDR Gram-negative infections [49].

As per individual antimicrobial regimens, quinolones-, β-lactam- and tetracycline-based therapy were significantly associated with lower crude mortality, whereas only tetracycline contributed to non-statistically significant shorter LOS with respect to median LOS. Other antimicrobial regimens were non-statistically related to relatively high crude, 14- and 28-day odds of mortality as well as longer LOS (>12 days). This contradiction is logically caused by the variation in the nature and severity of infections as well as the study population, highlighting the need to individualize the assessment of antimicrobial-related treatment outcomes for each infection type and patient group.

Finally, this analysis was retrospective and limited to the data already recorded in the patients’ electronic files which did not include genotyping of the patients nor classification based on disease severity. Future prospective studies should include patients’ genetic data to allow further stratification of patients for proper care and individualization of therapy.

## 5. Conclusions

G6PD patients are a unique population that is more vulnerable to infections. Consideration of the above risk factors should help practitioners gauge the intensity of antimicrobial therapy. While the comparison of combined versus monotherapy was not significant, the focus should be on individualized, targeted antibiotics. Finally, awareness of the special needs of this patient group may improve treatment outcomes.

## Figures and Tables

**Figure 1 antibiotics-12-00494-f001:**
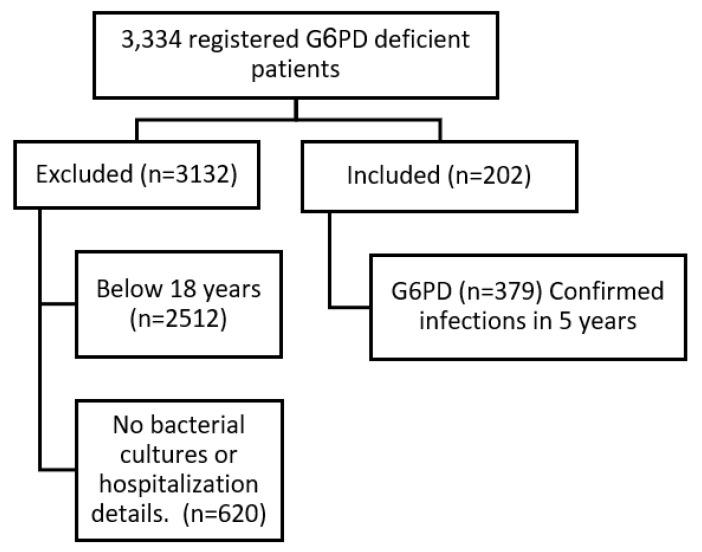
Study population: inclusion and exclusion.

**Table 1 antibiotics-12-00494-t001:** Patient’s Demographics.

	Overall N (%)	Recovery N (%)	Death N (%)	*p* Value Chi
Male	256 (68)	192 (70.1)	73 (69.5)	1.00
median (IQR)	60 (41–77)	53 (35–72)	77 (68–80)	<0.001
Age ≤ 60 Years	190 (50)	172 (62.8)	18 (17.1)	<0.001
Age > 60 years	189 (50)	102 (37.2)	87 (82.9)	
LOS median (IQR)	12 (5–31)	9 (5–24)	26 (13–43)	<0.001
LOS ≤ 14 days	195 (51)	168 (61.3)	27 (25.7)	<0.001
LOS > 14 days	184 (49)	106 (38.7)	78 (74.3)	
Admission with infectious disease	273 (72)	183 (66.8)	90 (85.7)	<0.001
Admission to critical care area	117 (31)	57 (20.8)	60 (57.1)	<0.001
**Underlying Comorbidities**				
Diabetes	254 (67)	170 (62.0)	84 (80.0)	0.001
Chronic kidney disease	228 (60)	141 (51.5)	87 (82.9)	<0.001
Active malignancy	14 (4)	7 (2.6)	7 (6.7)	0.111
Immunosuppressed	24 (6)	15 (5.5)	9 (8.6)	0.383
Chronic Cardiac Diseases	282 (74)	185 (67.5)	97 (92.4)	<0.001
HIV follow-up AIDS	1 (0)	0 (0.0)	1 (1.0)	0.618
Chronic Resp. Disease	70 (18)	27 (9.9)	43 (41.0)	<0.001
Sickle Cell	19 (5)	18 (6.6)	1 (1.0)	0.048
Other comorbidities	179 (47)	122 (44.5)	57 (54.3)	0.112
No. of comorbidities median (IQR)	3 (2–4)	4 (3–4)	3 (2–4)	<0.001
Any comorbidity	338 (89)	234 (85.4)	104 (99.0)	<0.001
**Risk Factors for infection**				
Blood transfusion during admission	212 (56)	128 (46.7)	84 (80.0)	<0.001
Invasive procedure during admission	285 (75)	182 (66.4)	103 (98.1)	<0.001
Surgery 90-day history	39 (10)	35 (12.8)	4 (3.8)	0.017
**Type of infection**				
Bacteremia	88 (23)	56 (20.4)	32 (30.5)	0.053
Body Fluids	1 (0)	1 (0.4)	0 (0.0)	1.000
Respiratory infections	91 (24)	53 (19.3)	38 (36.2)	0.001
Skin and soft tissue infections	103 (27)	86 (31.4)	17 (16.2)	0.004
Urinary tract infections	96 (25)	78 (28.5)	18 (17.1)	0.033
Gram-negative infections	227 (60)	163 (59.5)	64 (61.0)	0.886
Gram-positive infections	107 (28)	80 (29.2)	27 (25.7)	0.585
Fungal infections	31 (8)	22 (8.0)	9 (8.6)	1.000
SARS-CoV19 infections	14 (4)	9 (3.3)	5 (4.8)	0.705
CRE infections	24 (6)	21 (7.7)	3 (2.9)	0.138
ESBL infections	50 (13)	35 (12.8)	15 (14.3)	0.826
MDR infections	67 (18)	39 (14.2)	28 (26.7)	0.007
MRSA infections	12 (3)	11 (4.0)	1 (1.0)	0.232
Resistant phenotypes	153 (40)	106 (38.7)	47 (44.8)	0.336
Community-acquired infections	212 (56)	180 (65.7)	32 (30.5)	<0.001
Hospital-acquired infections	167 (44)	94 (34.3)	73 (69.5)	<0.001
90-day recurrence of any infection	136 (36)	131 (47.8)	5 (4.8)	<0.001
**Antimicrobial treatment**				
Cephalosporins (90-day exposure)	105 (28)	76 (27.7)	29 (27.6)	1.000
Aminoglycosides (90-day exposure)	25 (7)	21 (7.7)	4 (3.8)	0.262
Tetracyclines (90-day exposure)	19 (5)	15 (5.5)	4 (3.8)	0.688
B-lactam/B-lactamase (90-day exposure)	58 (15)	32 (11.7)	26 (24.8)	0.003
Macrolides (90-day exposure)	34 (9)	16 (5.8)	18 (17.1)	0.001
Glycopeptides (90-day exposure)	33 (9)	29 (10.6)	4 (3.8)	0.059
Nitroimidazole (90-day exposure)	29 (8)	18 (6.6)	11 (10.5)	0.287
Colistin (90-day exposure)	1 (0)	1 (0.4)	0 (0.0)	1.000
B-lactams (90-day exposure)	70 (18)	42 (15.3)	28 (26.7)	0.016
Glycylcycline (90-day exposure)	3 (1)	3 (1.1)	0 (0.0)	0.668
Quinolones (90-day exposure)	64 (17)	49 (17.9)	15 (14.3)	0.494
Oxazolidinones (90-day exposure)	2 (1)	2 (0.7)	0 (0.0)	0.932
90-day exposure to any antibiotic	158 (42)	108 (39.4)	50 (47.6)	0.182
**History of infection**				
90-days prior infection	87 (23)	63 (23.0)	24 (22.9)	1.000
Previous Gram-negative infection	48 (13)	41 (15.0)	7 (6.7)	0.045
Previous Gram-positive infection	34 (9)	32 (11.7)	2 (1.9)	0.005
Previous fungal infections	4 (1)	4 (1.5)	0 (0.0)	0.495
Previous SARS-CoV-19 infection	20 (5)	3 (1.1)	17 (16.2)	<0.001
Polymicrobial infections	224 (59)	137 (50.0)	87 (82.9)	<0.001
Concurrent Gram-negative infection	169 (45)	98 (35.8)	71 (67.6)	<0.001
Concurrent Gram-positive infection	106 (28)	68 (24.8)	38 (36.2)	0.038
Concurrent Fungal infection	59 (16)	25 (9.1)	34 (32.4)	<0.001
Concurrent SARS-CoV-19 infection	7 (2)	0 (0.0)	7 (6.7)	<0.001

Shading cells are the title for subsequent rows.

**Table 2 antibiotics-12-00494-t002:** Predictors for infection-related crude all-cause in-hospital mortality.

	Recovery *n* (%)	Death *n* (%)	Univariable OR (CI, *p*)		OR (Multivariable)
Age Mean (SD)	53.2 (21.2)	72.9 (12.7)	1.1 (1.0–1.1, *p* < 0.001)	#	1.13 (1.05–1.24, *p* = 0.003)
Age > 60 years	102 (54.0)	87 (46.0)	8.2 (4.7–14.7, *p* < 0.001)	#	3.00 (0.20–45.77, *p* = 0.426)
LOS Mean (SD)	17.7 (19.2)	46.4 (48.7)	1.03 (1.02–1.04, *p* < 0.001)	#	0.98 (0.96–1.02, *p* = 0.321)
LOS > 14 days	106 (57.6)	78 (42.4)	4.58 (2.81–7.66, *p* < 0.001)	#	0.05 (0.01–0.35, *p* = 0.004)
Admission with infectious disease	183 (67.0)	90 (33.0)	2.98 (1.68–5.63, *p* < 0.001)	#	14.59 (2.0–156.98, *p* = 0.015)
Admission to critical care area	57 (48.7)	60 (51.3)	5.08 (3.14–8.29, *p* < 0.001)	#	46.39 (5.1–716.25, *p* = 0.002)
Diabetes	170 (66.9)	84 (33.1)	2.45 (1.45–4.27, *p* = 0.001)	#	0.03 (0.00–0.48, *p* = 0.077)
Chronic kidney disease	141 (61.8)	87 (38.2)	4.56 (2.66–8.19, *p* < 0.001)	#	0.12 (0.00–2.51, *p* = 0.328)
Active malignancy	7 (50.0)	7 (50.0)	2.72 (0.91–8.15, *p* = 0.067)	#	0.11 (0.00–71.31, *p* = 0.598)
Chronic Cardiac Diseases	185 (65.6)	97 (34.4)	5.83 (2.87–13.5, *p* < 0.001)	#	0.15 (0.00–6.99, *p* = 0.423)
Chronic Resp. Disease	27 (38.6)	43 (61.4)	6.34 (3.66–11.2, *p* < 0.001)	#	1.10 (0.00–29.04, *p* = 0.966)
Sickle Cell	18 (94.7)	1 (5.3)	0.14 (0.01–0.68, *p* = 0.054)	#	0.01 (0.00–0.87, *p* = 0.081)
Other comorbidities	122 (68.2)	57 (31.8)	1.48 (0.94–2.33, *p* = 0.089)	#	0.36 (0.00–7.60, *p* = 0.626)
Comorbid Mean (SD)	3.5 (1.4)	4.7 (1.1)	2.06 (1.67–2.58, *p* < 0.001)	#	6.11 (0.36–2119.6, *p* = 0.358)
Any comorbidity	234 (69.2)	104 (30.8)	17.8 (3.8–317.5, *p* = 0.005)	#	111.3 (1.3–15,131, *p* = 0.045)
Blood transfusion during admission	128 (60.4)	84 (39.6)	4.56 (2.72–7.94, *p* < 0.001)	#	0.95 (0.21–4.00, *p* = 0.942)
Invasive procedure during admission	182 (63.9)	103 (36.1)	26.03 (8–160.03, *p* < 0.001)	#	392.6 (37.8–7562, *p* < 0.001)
Surgery 90-day history	35 (89.7)	4 (10.3)	0.27 (0.08–0.70, *p* = 0.016)	#	0.11 (0.01–0.73, *p* = 0.031)
Bacteremia	56 (63.6)	32 (36.4)	1.71 (1.02–2.83, *p* = 0.040)	#	* *p* = 0.992
Respiratory infections	53 (58.2)	38 (41.8)	2.36 (1.43–3.89, *p* = 0.001)	#	* *p* = 0.992
Skin and soft tissue infections	86 (83.5)	17 (16.5)	0.42 (0.23–0.74, *p* = 0.003)	#	* *p* = 0.993
Urinary tract infections	78 (81.2)	18 (18.8)	0.52 (0.29, 0.92, *p* = 0.025)	#	* *p* = 0.993
CRE infections	21 (87.5)	3 (12.5)	0.35 (0.08–1.06, *p* = 0.099)	#	0.02 (0.00–0.39, *p* = 0.013)
MDR infections	39 (58.2)	28 (41.8)	2.19 (1.26–3.79, *p* = 0.005)	#	0.64 (0.13–3.11, *p* = 0.575)
MRSA infections	11 (91.7)	1 (8.3)	0.23 (0.01–1.20, *p* = 0.162)	#	1.36 (0.04–19.46, *p* = 0.840)
Hospital-acquired infections	94 (56.3)	73 (43.7)	4.37 (2.71–7.17, *p* < 0.001)	#	5.22 (1.13–28.78, *p* = 0.043)
Aminoglycosides (90-day exposure)	21 (84.0)	4 (16.0)	0.48 (0.14–1.29, *p* = 0.185)	#	2.17 (0.02–210.65, *p* = 0.739)
B-lactam/B-lactamase (90-day exposure)	32 (55.2)	26 (44.8)	2.49 (1.39–4.43, *p* = 0.002)	#	141.9 (10.7–2759, *p* < 0.001)
Macrolides (90-day exposure)	16 (47.1)	18 (52.9)	3.34 (1.63–6.89, *p* = 0.001)	#	1.68 (0.09–31.01, *p* = 0.724)
Glycopeptides (90-day exposure)	29 (87.9)	4 (12.1)	0.33 (0.10–0.88, *p* = 0.045)	#	0.07 (0.00–2.54, *p* = 0.171)
Nitroimidazole (90-day exposure)	18 (62.1)	11 (37.9)	1.66 (0.74–3.61, *p* = 0.204)	#	39.83 (1.2–1506.5, *p* = 0.043)
B-lactams (90-day exposure)	42 (60.0)	28 (40.0)	2.01 (1.16–3.45, *p* = 0.012)	#	15.13 (1.57–172.3, *p* = 0.021)
90-day exposure to any antibiotic	108 (68.4)	50 (31.6)	1.40 (0.89–2.20, *p* = 0.148)	#	0.28 (0.02–3.55, *p* = 0.330)
Previous Gram-negative infection	41 (85.4)	7 (14.6)	0.41 (0.16–0.88, *p* = 0.034)	#	0.72 (0.04–9.96, *p* = 0.812)
Previous Gram-positive infection	32 (94.1)	2 (5.9)	0.15 (0.02–0.50, *p* = 0.009)	#	0.01 (0.00–0.68, *p* = 0.052)
Previous SARSCoV19 infection	3 (15.0)	17 (85.0)	17.45 (5.70–76, *p* < 0.001)	#	1.42 (0.01–123.01, *p* = 0.873)
Polymicrobial infections	137 (61.2)	87 (38.8)	4.83 (2.82–8.69, *p* < 0.001)	#	1.91 (0.19–19.60, *p* = 0.581)
Concurrent Gram-negative infection	98 (58.0)	71 (42.0)	3.75 (2.34–6.10, *p* < 0.001)	#	3.03 (0.51–21.84, *p* = 0.241)
Concurrent Gram-positive infection	68 (64.2)	38 (35.8)	1.72 (1.06–2.78, *p* = 0.028)	#	0.71 (0.15–3.24, *p* = 0.652)
Concurrent Fungal infection	25 (42.4)	34 (57.6)	4.77 (2.68–8.59, *p* < 0.001)	#	1.73 (0.25–12.42, *p* = 0.575)
Concurrent SARSCoV19 infection	0 (0.0)	7 (100.0)	* *p* = 0.976		

#: variables with *p*-values ≤ 0.2 are enrolled in multivariate analysis, *: value is too high/too low to be detected by the software, 95% CI: Confidence intervals.

**Table 3 antibiotics-12-00494-t003:** Predictors for infection-related 14-day mortality.

	No	Yes	Univariable OR (CI, *p*)		Multivariable OR (CI, *p*)
Male	241 (90.9)	24 (9.1)	3.68 (1.25–15.73, *p* = 0.036)	#	5.01 (0.70–65.98, *p* = 0.154)
Age Mean (SD)	57.5 (21.1)	73.4 (16.0)	1.05 (1.02–1.07, *p* < 0.001)	#	1.06 (0.99–1.14, *p* = 0.116)
Age > 60 years	167 (88.4)	22 (11.6)	4.87 (1.95–14.81, *p* = 0.002)	#	1.06 (0.08–15.47, *p* = 0.967)
Length of stay mean (SD)	27.2 (33.7)	5.3 (3.7)	0.85 (0.77–0.92, *p* < 0.001)	#	0.71 (0.58–0.83, *p* < 0.001)
Chronic kidney disease	208 (91.2)	20 (8.8)	1.98 (0.85–5.15, *p* = 0.132)	#	0.60 (0.11–3.20, *p* = 0.550)
Blood transfusion during admission	201 (94.8)	11 (5.2)	0.52 (0.23–1.14, *p* = 0.104)	#	1.73 (0.43–7.20, *p* = 0.441)
Invasive procedure during admission	260 (91.2)	25 (8.8)	4.42 (1.28–27.83, *p* = 0.046)	#	100.8 (13.88–1455.8, *p* < 0.001)
Bacteremia	75 (85.2)	13 (14.8)	3.43 (1.53–7.65, *p* = 0.002)	#	2.06 (0.40–11.46, *p* = 0.389)
Skin and soft tissue infections	99 (96.1)	4 (3.9)	0.44 (0.13–1.19, *p* = 0.144)	#	0.40 (0.06–2.50, *p* = 0.333)
Gram-negative infections	217 (95.6)	10 (4.4)	0.37 (0.16–0.81, *p* = 0.015)	#	0.24 (0.03–1.99, *p* = 0.172)
Gram-positive infections	94 (87.9)	13 (12.1)	2.55 (1.14–5.65, *p* = 0.020)	#	1.00 (0.11–9.98, *p* = 0.998)
Community-acquired infections	162 (97.0)	5 (3.0)	3.9 (0.09–0.67, *p* = 0.009)	#	4.80 (0.57–50.13, *p* = 0.162)
Aminoglycosides (90-day exposure)	21 (84.0)	4 (16.0)	2.74 (0.75–7.96, *p* = 0.086)	#	189.14 (2.47–18,342.45, *p* = 0.020)
Tetracyclines (90-day exposure)	16 (84.2)	3 (15.8)	2.62 (0.58–8.58, *p* = 0.146)	#	0.35 (0.01–14.33, *p* = 0.559)
B-lactam/B-lactamase (90-day exposure)	57 (98.3)	1 (1.7)	0.20 (0.01–0.97, *p* = 0.117)	#	0.01 (0.00–0.23, *p* = 0.013)
90-day exposure to any antibiotic	151 (95.6)	7 (4.4)	0.47 (0.18–1.08, *p* = 0.091)	#	0.28 (0.04–1.46, *p* = 0.157)
90-days prior infection	84 (96.6)	3 (3.4)	0.40 (0.09–1.18, *p* = 0.141)	#	0.21 (0.02–1.70, *p* = 0.179)
Polymicrobial infections	212 (94.6)	12 (5.4)	0.53 (0.24–1.16, *p* = 0.113)	#	2.68 (0.24–24.41, *p* = 0.386)
Concurrent Gram-negative infection	161 (95.3)	8 (4.7)	0.50 (0.20–1.13, *p* = 0.110)	#	0.36 (0.04–4.35, *p* = 0.394)
Concurrent SARSCoV19 infection	4 (57.1)	3 (42.9)	10.87 (2.05–52.10, *p* = 0.003)	#	4.92 (0.04–1209.39, *p* = 0.612)

#: variables with *p*-values ≤ 0.2 are enrolled in multivariate analysis, 95% CI: Confidence intervals.

**Table 4 antibiotics-12-00494-t004:** Predictors for infection-related 28-day mortality *.

	No	Yes	Univariable OR (CI, *p*)		Multivariable OR (CI, *p*)
Age Mean (SD)	57.6 (21.3)	70.8 (14.8)	1.04 (1.01–1.06, *p* = 0.002)	#	0.96 (0.89–1.03, *p* = 0.257)
Age > 60 years	164 (86.8)	25 (13.2)	5.64 (2.29–17.01, *p* = 0.001)	#	66.81 (3.58–2143.23, *p* = 0.009)
Admission with infectious disease	247 (90.5)	26 (9.5)	2.68 (1.01–9.27, *p* = 0.073)	#	4.02 (0.97–21.13, *p* = 0.072)
Chronic kidney disease	205 (89.9)	23 (10.1)	2.31 (1.01–5.95, *p* = 0.060)	#	0.02 (0.00–0.40, *p* = 0.015)
Active malignancy	8 (57.1)	6 (42.9)	10.66 (3.28–33.21, *p* < 0.001)	#	* (*p* = 0.830)
Immunosuppressed	18 (75.0)	6 (25.0)	4.60 (1.55–12.15, *p* = 0.003)	#	0.00 (0.00–1.13, *p* = 0.843)
Chronic Cardiac Diseases	255 (90.4)	27 (9.6)	3.32 (1.14–14.12, *p* = 0.053)	#	4.30 (0.39–67.64, *p* = 0.260)
Chronic Resp. Disease	68 (97.1)	2 (2.9)	0.30 (0.05–1.02, *p* = 0.101)	#	0.00 (0.00–0.07, *p* = 0.001)
Other comorbidities	155 (86.6)	24 (13.4)	5.01 (2.12–13.79, *p* = 0.001)	#	0.13 (0.01–1.51, *p* = 0.117)
No. of comorbid Mean (SD)	3.7 (1.4)	4.8 (1.6)	1.81 (1.33–2.55, *p* < 0.001)	#	3.16 (0.77–15.18, *p* = 0.123)
Blood transfusion	187 (88.2)	25 (11.8)	4.33 (1.75–13.06, *p* = 0.003)	#	13.70 (2.85–91.25, *p* = 0.003)
Respiratory infections	79 (86.8)	12 (13.2)	2.28 (1.03–4.89, *p* = 0.037)	#	1.28 (0.38–4.35, *p* = 0.689)
Skin and soft tissue infections	102 (99.0)	1 (1.0)	0.08 (0.00–0.40, *p* = 0.015)	#	0.14 (0.00–1.35, *p* = 0.148)
Fungal infections	26 (83.9)	5 (16.1)	2.48 (0.79–6.58, *p* = 0.086)	#	4.35 (0.54–34.25, *p* = 0.157)
ESBL infections	49 (98.0)	1 (2.0)	0.21 (0.01–1.02, *p* = 0.131)	#	0.38 (0.02–3.09, *p* = 0.431)
Place of Acquisition	145 (86.8)	22 (13.2)	3.87 (1.74–9.48, *p* = 0.002)	#	6.62 (1.44–37.62, *p* = 0.021)
Cephalosporins (90-day exposure)	92 (87.6)	13 (12.4)	2.14 (0.98–4.55, *p* = 0.050)	#	13.21 (2.33–102.04, *p* = 0.007)
B-lactam/B-lactamase (90-day exposure)	49 (84.5)	9 (15.5)	2.62 (1.09–5.91, *p* = 0.024)	#	22.11 (4.66–137.65, *p* < 0.001)
Polymicrobial Infections	196 (87.5)	28 (12.5)	10.93 (3.22–68.35, *p* = 0.001)	#	31.80 (1.97–934.33, *p* = 0.024)
Concurrent Gram-negative infection	144 (85.2)	25 (14.8)	7.12 (2.88–21.48, *p* < 0.001)	#	4.73 (0.97–34.53, *p* = 0.081)

#: variables with *p*-values ≤ 0.2 are enrolled in multivariate analysis, *: value is too high/too low to be detected by the software, 95% CI: Confidence intervals.

**Table 5 antibiotics-12-00494-t005:** Treatment outcomes Vs. Antimicrobial treatment regimen (Binary logistic regression).

Antimicrobial Therapy	Overall	14-Day Mortality (27)			28-Day Mortality (30)			All-Cause In-Hospital Mortality (105)			LOS			
	no. (%)	no. (%)	*p*	OR	no. (%)	*p*	OR	no. (%)	*p*	OR	Median	IQR	*p*	OR
Combined	129 (34)	11 (41)	0.45	1.4	7 (23)	0.20	0.6	35 (33)	0.86	1.0	11	(6–27)	0.21	1.0
Monotherapy	250 (66)	16 (59)	0.45	0.7	23 (77)	0.20	1.8	70 (67)	0.86	1.0	16	(5–34)	0.21	1.0
Cephalosporin-based	157 (41)	16 (59)	0.06	2.2	10 (33)	0.35	0.7	51 (49)	0.08	1.5	9	(5–23)	0.35	1.0
β-lactam/β-lactamase inhibitor-based	131 (35)	13 (48)	0.13	1.8	13 (43)	0.30	1.5	46 (44)	0.02	1.7	16	(5–38)	0.10	1.0
Piperacillin/Tazobactam based	99 (26)	9 (33)	0.38	1.5	9 (30)	0.62	1.2	36 (34)	0.03	1.7	24	(8–50)	0.00	1.0
Quinolones-based	45 (12)	0 (0)	0.97	*	1 (3)	0.17	0.2	2 (2)	0.00	0.1	16	(9–25)	0.37	1.0
Vancomycin-based	38 (10)	1 (4)	0.28	0.3	6 (20)	0.07	2.5	10 (10)	0.84	0.9	11	(5–26)	0.07	1.0
β-lactam-based	37 (10	0 (0)	0.96	*	3 (10)	0.96	1.0	5 (5)	0.05	0.4	18	(8–41)	0.99	1.0
Antifungal	31 (8)	1 (4)	0.39	0.4	5 (17)	0.09	2.5	9 (9)	0.86	1.1	16	(5–40)	0.91	1.0
Tetracycline-based	30 (8)	2 (7)	0.92	0.9	0 (0)	0.96	*	2 (2)	0.02	0.2	6.5	(5–19)	0.38	1.0
Meropenem based	28 (7)	0 (0)	0.96	*	3 (10)	0.57	1.4	4 (4)	0.11	0.4	18	(10–41)	0.49	1.0
Macrolide-based	28 (7)	6 (22)	0.01	4.3	3 (10)	0.57	1.4	12 (11)	0.07	2.1	9	(4–17)	0.12	1.0
Colistin-based	24 (6)	0 (0)	0.97	*	4 (13)	0.11	2.5	7 (7)	0.87	1.1	40	(22–60)	0.01	1.0
Aminoglycosides-based	22 (6)	0 (0)	0.97	*	0 (0)	0.97	*	7 (7)	0.66	1.2	11	(7–37)	0.72	1.0

* Value is too low to be detected by software.

## Data Availability

Long tables with detailed information in the Appendix A. Raw data are available on request, binding approval from local authorities and signature of proper documentation.

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
