# Peer review of "Infections in Glucose-6-Phosphate Dehydrogenase G6PD-Deficient Patients; Predictors for Infection-Related Mortalities and Treatment Outcomes"

_antibiotics, 2023, doi:10.3390/antibiotics12030494_

Round 1
Reviewer 1 Report
This study has analyzed the predictors of infection-related mortalities in G6PD deficient patients, and found that age and some factors are related to mortality. Authors also claimed that awareness of 36 the special needs of this patient group may improve treatment outcomes.
However, I have some concerns about the manuscript.
1) I have found an article published by the same group, “Alrahmany, D., Omar, A. F., Al-Maqbali, S. R. S., Harb, G., & Ghazi, I. M. (2022). Infections in G6PD-Deficient Hospitalized Patients-Prevalence, Risk Factors, and Related Mortality. Antibiotics (Basel, Switzerland), 11(7), 934. https://doi.org/10.3390/antibiotics11070934”. This manuscript has the same study population with the published article (the methods are same, and the included patients are some). Some findings of this submission, is already published by the former article.
2) Although many factors are related to mortality, the causality is unsure. For example, exposure to multi-class antimicrobials, it is very possible that patients are severe illness and have to receive antimicrobials. Some factors, such as age, are usual and obvious.
3) Many grammar errors exist. Figure 1 is garbled. Legend of table in section “3.4.1” is missing.
Author Response
This study has analyzed the predictors of infection-related mortalities in G6PD deficient patients and found that age and some factors are related to mortality. Authors also claimed that awareness of 36 the special needs of this patient group may improve treatment outcomes.
However, I have some concerns about the manuscript.
- I have found an article published by the same group, “Alrahmany, D., Omar, A. F., Al-Maqbali, S. R. S., Harb, G., & Ghazi, I. M. (2022). Infections in G6PD-Deficient Hospitalized Patients-Prevalence, Risk Factors, and Related Mortality. Antibiotics (Basel, Switzerland), 11(7), 934. https://doi.org/10.3390/antibiotics11070934”. This manuscript has the same study population with the published article (the methods are same, and the included patients are some). Some findings of this submission, is already published by the former article.
Response: Thank you for the opportunity to improve the manuscript. While our previous publication is concerned with the same disease state, the two manuscripts focus on and discuss different and distinct aspects of the topic. In the first manuscript, we comprehensively described the type of infections, the causative organisms, resistance phenotypes, and antimicrobial susceptibility patterns. Furthermore, we described the infection-acquisition aspects, with a close focus on MDR-acquired infections and their related risk factors, additionally, we identified the pattern and risk factors for hospital-acquired infections.
In the current article, we studied crude, 14-day, and 28-day mortality. An exhaustive list of potential risk factors was included in binary and multivariate regression analysis in addition to correlation testing. Furthermore, we discussed the clinical background of each risk factor and the opportunities for either tailored clinical or antimicrobial stewardship interventions to improve clinical outcomes and reduce mortality.
We also studied the treatment outcomes of commonly used antimicrobial regimens regarding mortality and LOS.
- Although many factors are related to mortality, the causality is unsure. For example, exposure to multi-class antimicrobials, it is very possible that patients are severe illness and have to receive antimicrobials. Some factors, such as age, are usual and obvious.
Response: Thank you for the comment. The scope of the study is to analyze risk factors statistically to demonstrate significance by regression analysis on a predictive, not explanatory basis. The mechanistic causality is important to corroborate the findings and could be the topic of future studies. (Schooling CM, Jones HE. Clarifying questions about "risk factors": predictors versus explanation. Emerg Themes Epidemiol. 2018 Aug 8;15:10. doi: 10.1186/s12982-018-0080-z. PMID: 30116285; PMCID: PMC6083579)
- Many grammar errors exist. Figure 1 is garbled. Legend of table in section “3.4.1” is missing.
Response: Thank you for the remark. Modifications are made in the text. The manuscript has been extensively reviewed by language software and two independent native-speaker college professors.
Reviewer 2 Report
This paper deals with the problem related to the deficiency of Glucose-6-phosphate dehydrogenase (G6PD) which is an essential enzyme for the maintenance of blood cell integrity leading to premature loss of celularl integrity suggesting that the G6PD deficiency could be a significant predictor of hospitalization and severe infections.
The non-specific innate immune response achieves integrated, highly specialized functions that permit to combact the infections.
The G6PD enzyme deficiency is a genetic blood diseases that impacts the body’s ability to resist to bacterial or viral infections so that a prompt and appropriate antimicrobial therapy should be warranted to combat infections and to cover the special needs of this patient group.
Given the above. I wonder whether this article might be interesting and useful for a general audience due to the fact that this genetic disease only concerns a very small part of the population. Just in authors community -due to the high prevalence of consanguineous marriage- and in a few other parts of the world, a certain number of G6PD enzyme deficiency cases is reported (see Oman, Arabia, Yemen) etc). Overcoming this issue, the authors collected a quite significant amount of data from their country over a five-years period including G6PD-deficient genetically tested adult patients (>18 years) with laboratory confirmed microbial infections (totally 202 patients and 379 microbiological cultures) correlating this data with many parameters such as risk factors, co-morbidities, previous antibiotic therapies and so on. However at this point I noticed that the data are shown in an unclear and confusing way. For instance, the 379 microbiological data are not clearly explained and there are no correspondence between the numbers reported in the tables. We know that only the first episode was selected for patients with identical cultures and the same antibiotics pattern so that identical cultures isolated within 30 days for the same patient are considered unique. What happened for other cultures? The numbers do not correspond in the tables (for example admission details, co-morbidities ets outnumber the 379 ). Please give me an explanation about the structuring of the tables. Some minor English errors have to be corrected (see lines 113, 114,121 etc) as well as the bacteria and yeast names (Coagulase-negative Staphylococcus and candida, lines 338-339) should be written in italics.
Moreover the authors should specify where exactly their tertiary care hospital is located.
Author Response
- This paper deals with the problem related to the deficiency of Glucose-6-phosphate dehydrogenase (G6PD) which is an essential enzyme for the maintenance of blood cell integrity leading to premature loss of celularl integrity suggesting that the G6PD deficiency could be a significant predictor of hospitalization and severe infections.
The non-specific innate immune response achieves integrated, highly specialized functions that permit to combact the infections.
The G6PD enzyme deficiency is a genetic blood diseases that impacts the body’s ability to resist to bacterial or viral infections so that a prompt and appropriate antimicrobial therapy should be warranted to combat infections and to cover the special needs of this patient group.
Given the above. I wonder whether this article might be interesting and useful for a general audience due to the fact that this genetic disease only concerns a very small part of the population. Just in authors community -due to the high prevalence of consanguineous marriage- and in a few other parts of the world, a certain number of G6PD enzyme deficiency cases is reported (see Oman, Arabia, Yemen) etc). Overcoming this issue, the authors collected a quite significant amount of data from their country over a five-years period including G6PD-deficient genetically tested adult patients (>18 years) with laboratory confirmed microbial infections (totally 202 patients and 379 microbiological cultures) correlating this data with many parameters such as risk factors, co-morbidities, previous antibiotic therapies and so on.
Response: Thank you for the comment. The global prevalence of G6PD deficiency is around 400 million individuals (Nkhoma ET, Poole C, Vannappagari V, Hall SA, Beutler E. The global prevalence of glucose-6-phosphate dehydrogenase deficiency: a systematic review and meta-analysis. Blood Cells Mol Dis. 2009 May-Jun;42(3):267-78. doi: 10.1016/j.bcmd.2008.12.005. Epub 2009 Feb 23. PMID: 19233695.).
The existence of cluster cases in our community presents a unique opportunity to study a number of cases large enough to generate more meaningful data.
- However, at this point, I noticed that the data are shown in an unclear and confusing way. For instance, the 379 microbiological data are not clearly explained and there are no correspondence between the numbers reported in the tables.
Response: Thank you for the comment. Table 1 describes an exhaustive list of demographics related to patients who are linked to 379 microbiological cultures. Tables 2,3 and 4 describe potential risk factors for crude, 14-day, and 28-day mortality-based results from binary, multiple, and correlation statistics. We have added some clarifications to the tables.
- We know that only the first episode was selected for patients with identical cultures and the same antibiotics pattern so that identical cultures isolated within 30 days for the same patient are considered unique. What happened for other cultures?
Response: Thank you for the comment. If any cultures were collected and considered redundant as per the study definition mentioned above, the cultures were excluded from the analysis because they did not represent a unique event
- The numbers do not correspond in the tables (for example admission details, co-morbidities ets outnumber the 379 ). Please give me an explanation about the structuring of the tables.
Response: The tables have been reviewed. Any mutually exclusive events add up to the total number of inclusion (men vs women, admitted to ICU vs general ward) however other events like (cardiac condition and respiratory conditions) can’t add up because of the possibility of a patient having either or both at the same time.
- Some minor English errors have to be corrected (see lines 113, 114,121 etc) as well as the bacteria and yeast names (Coagulase-negative Staphylococcus and candida, lines 338-339) should be written in italics.
Response: Thank you for your comment. The manuscript has been extensively reviewed. Any genus and species names are italicized.
- Moreover the authors should specify where exactly their tertiary care hospital is located.
Response: Our hospital is located in Suhar province, Al Batinah governorate, Oman. Added to the manuscript.
Reviewer 3 Report
Alrahmany et al. performed a retrospective polycentric study in Oman between 2017 and 2021 in G6PD deficient patients. It is of interest to demonstrate that many risk factors were contributed significantly to deaths, as some authors could have suggested it.
・Nevertheless, the authors conclude that strict application of stewardship principles may be critical for G6PD patients. The authors cannot conclude them. This conclusion would be appropriate if a control group was available which is not the case.
・Another comment is the very high mortality in patients with G6PD. The population needs then to be better described, for example with subcomponents of SOFA scores. The expected mortality would be around 20-30 %.
・Also from a methodological point of view a main outcome should be clearly stated as well as a main analysis and a necessary number of patients to be enrolled calculated.
・I would suggest to present a table with the outcomes, such as use of catecholamines, use of artificial ventilation.
Author Response
Alrahmany et al. performed a retrospective polycentric study in Oman between 2017 and 2021 in G6PD deficient patients. It is of interest to demonstrate that many risk factors were contributed significantly to deaths, as some authors could have suggested it.
- Nevertheless, the authors conclude that strict application of stewardship principles may be critical for G6PD patients. The authors cannot conclude them. This conclusion would be appropriate if a control group was available which is not the case.
Response: The authors stated “maybe” signifying the lack of head-to-head comparison of prospective data to confirm the impact application of these measures. The language was modified to “beneficial”.
- Another comment is the very high mortality in patients with G6PD. The population needs then to be better described, for example with subcomponents of SOFA scores. The expected mortality would be around 20-30 %.
Response: Thank you for the comment. Unfortunately, such information was not initially collected.
- Also from a methodological point of view, a main outcome should be clearly stated as well as a main analysis and a necessary number of patients to be enrolled calculated.
Response: Thank you for the comment. The main outcomes are stated in the last part of the introduction (Lines 81-86). The sample size calculation was not warranted in our study as we analyzed all patients that matched the inclusion and exclusion criteria (convenience sample) in a retrospective manner.
- I would suggest to present a table with the outcomes, such as use of catecholamines, use of artificial ventilation.
Response: Thank you for the comment. The ultimate outcome that was the focus of the study was mortality, other secondary outcomes would have been useful as well, unfortunately, data regarding the catecholamines use and artificial ventilation were not collected.
Round 2
Reviewer 1 Report
The concerns are not well addressed.
Author Response
The comments of the reviewer remain unchanged from previous cycle, as per the editor's instruction no further response is required.
Reviewer 2 Report
Unfortunately the authors did not address correctly the reviwer’s comments.
Other issues are the following:
-The tables remain confusing and of difficult interpratation
-In table 1 the p-value between the dead and the recovered patints in order to evaluate the different statistical significance, is not reported.
- For tables 2,3,4, I suggest the authors ,for greater clarity of the tables, to only include the statistically significant variables in univariate and multivariate analyses. The multivariate analyses should also include the OR and CI values
-Table 5 must be semplified
-Table 6 should be removed . I am doubtful about the Spearman Correlation coefficients that should be used only between 2 quantitatve data.
-To my comment 3 in the first revision, the authors did not reply ia an appropriate way
-Why did the authors consider the mortality at 14-day and 28 -day mortality?,
Just in case these data should be added as supplementary material
Author Response
Reviewer 2 (2nd Cycle)
Unfortunately the authors did not address correctly the reviwer’s comments, Other issues are the following:
- The tables remain confusing and of difficult interpretation
Response:
- All tables are summarized to comprise only the statistically significant variables as per the reviewer's recommendation.
- All tables are restructured to be more clear for the reader.
- Short tables with only significant variables are added to the text, while comprehensive tables are added as an appendix.
- In table 1 the p-value between the dead and the recovered patints in order to evaluate the different statistical significance, is not reported.
Response
- A column containing all P values (Chi-square test) between variables in both groups, is added to table 1
- For tables 2,3,4, I suggest the authors ,for greater clarity of the tables, to only include the statistically significant variables in univariate and multivariate analyses. The multivariate analyses should also include the OR and CI values
Response:
- All tables are restructured to be more clear for the reader.
- Short tables with only significant variables are added to the text, while comprehensive tables are added as an appendix.
- ORs and CIs are added to the multivariate analysis
- Table 5 must be simplified
Response:
- The table is edited to be simple for the reader
- Table 6 should be removed . I am doubtful about the Spearman Correlation coefficients that should be used only between 2 quantitatve data.
Response:
- Thank you for your comment, Table 6 is deleted and the correlation coefficient column is deleted from all tables.
- To my comment 3 in the first revision, the authors did not reply ia an appropriate way (We know that only the first episode was selected for patients with identical cultures and the same antibiotics pattern so that identical cultures isolated within 30 days for the same patient are considered unique. What happened for other cultures?)
Response
- Thank you for your comment, redundant cultures were not included in the analysis to avoid skewing the data in any direction.
- Why did the authors consider the mortality at 14-day and 28 -day mortality?
Response
- Mortality at 14 and 28 days are the most common endpoints in infectious diseases literature representing response to treatment initiailly then detecting if any relapse of development of resistance. The authors have chosen hese endpoints to consistent with literature and facilitate comparisons.
- Just in case these data should be added as supplementary material
Response:
As per local rules and regualtions in effect in Sultanate of Oman, raw data sheets are available upon written request, MOH approval and signature of required paperwork.
Round 3
Reviewer 2 Report
None
Author Response
There is no further comments from the reviewer. Thank you for your guidance so far.